# A Plug-and-Play Method for Rare Human-Object Interactions Detection by Bridging Domain Gap

Lijun Zhang*
School of Computer Science and Ningbo Institute,
Northwestern Polytechnical University
Xi'an, Shaanxi, China
National Engineering Laboratory for Integrated
Aero-Space-Ground-Ocean
Xi'an, Shaanxi, China
lijunzhang@mail.nwpu.edu.cn

Wei Suo*
School of Computer Science and Ningbo Institute,
Northwestern Polytechnical University
Xi'an, Shaanxi, China
National Engineering Laboratory for Integrated
Aero-Space-Ground-Ocean
Xi'an, Shaanxi, China
suowei1994@mail.nwpu.edu.cn

Peng Wang†
School of Computer Science and Ningbo Institute,
Northwestern Polytechnical University
Xi'an, Shaanxi, China
National Engineering Laboratory for Integrated
Aero-Space-Ground-Ocean
Xi'an, Shaanxi, China
peng.wang@nwpu.edu.cn

Yanning Zhang
School of Computer Science and Ningbo Institute,
Northwestern Polytechnical University
Xi'an, Shaanxi, China
National Engineering Laboratory for Integrated
Aero-Space-Ground-Ocean
Xi'an, Shaanxi, China
ynzhang@nwpu.edu.cn

## Abstract

Human-object interactions (HOI) detection aims at capturing human-object pairs in images and corresponding actions. It is an important step toward high-level visual reasoning and scene understanding. However, due to the natural bias from the real world, existing methods mostly struggle with rare human-object pairs and lead to suboptimal results. Recently, with the development of the generative model, a straightforward approach is to construct a more balanced dataset based on a group of supplementary samples. Unfortunately, there is a significant domain gap between the generated data and the original data, and simply merging the generated images into the original dataset cannot significantly boost the performance. To alleviate the above problem, we present a novel model-agnostic framework called **C**ontext-**E**nhanced **F**eature **A**lignment (CEFA) module, which can effectively align the generated data with the original data at the feature level and bridge the domain gap. Specifically, CEFA consists of a feature alignment module and a context enhancement module. On one hand, considering the crucial role of human-object pairs information in HOI tasks, the feature alignment module aligns the human-object pairs by aggregating instance information. On the other hand, to mitigate the issue of losing important context information caused by the traditional discriminator-style alignment method, we employ a context-enhanced image reconstruction module to improve the model's learning ability of contextual cues. Extensive experiments have shown that our method can serve as a plug-and-play module to improve the detection performance of HOI models on rare categories[1].

## CCS Concepts

• **Computing methodologies** → **Computer vision**.

## Keywords

Human-Object Interactions, Domain Gap, Feature Alignment, Diffusion Module

**ACM Reference Format:**
Lijun Zhang, Wei Suo, Peng Wang, and Yanning Zhang. 2024. A Plug-and-Play Method for Rare Human-Object Interactions Detection by Bridging Domain Gap. In *Proceedings of the 32nd ACM International Conference on Multimedia (MM '24), October 28-November 1, 2024, Melbourne, VIC, Australia.* ACM, New York, NY, USA, 10 pages. https://doi.org/10.1145/3664647.3680666

## 1 Introduction

The goal of Human-Object Interaction (HOI) detection is to localize humans and objects in an image and recognize their interactions. The interactions can be represented by a triplet <Human, Verb, Object>. HOI enables machines to understand human activities at a fine-grained level, which is an important step towards high-level advanced visual reasoning [16, 28, 39] and scene understanding [29, 37, 41]. However, real-world data have a long-tailed distribution such as human riding giraffe, which will lead to the presence of long-tail phenomena in HOI datasets. Long-tail datasets result in mediocre performance of HOI models on rare categories.

---

*Both authors contributed equally to this research.
†Corresponding author

[1] https://github.com/LijunZhang01/CEFA

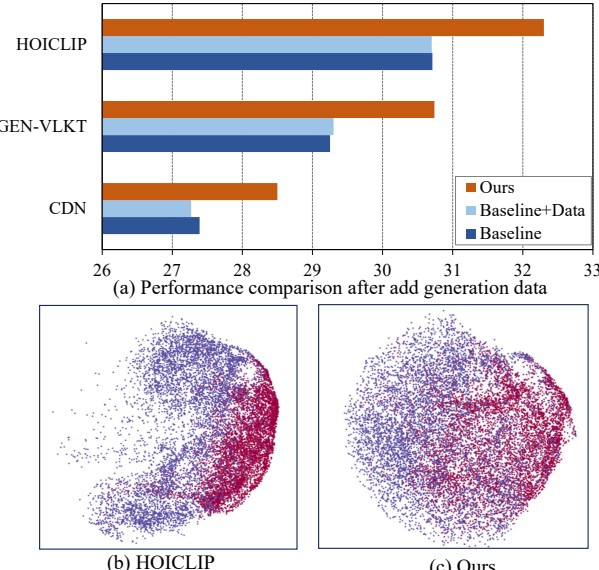

(a) Performance comparison after add generation data

(b) HOICLIP          (c) Ours

Figure 1: (a) Performance comparison of rare category on HICO-Det dataset. "Ours" refers to our CEFA method. "Baseline" represents traditional HOI models such as CDN, GEN-VLKT and HOICLIP. "Baseline+Date" indicates the approach of merely merging the generated data with the original dataset to train the baseline model. (b) Visualization of original image features and generated image features extracted using HOICLIP encoder, represented by t-SNE plots. Red represents the generated images, blue represents the original images. (c) Visualization of features extracted by our model.

To address the long-tail problem, existing methods can be divided into two paradigms [1, 13, 44]. The first category is based on training strategies [11, 33, 44], such as resampling and retraining techniques. Resampling and retraining are achieved by changing the sampling strategy for long-tail data or adjusting the loss function. The other research line focused on combinatorial learning [1, 13, 55]. They utilize the concept of combinatorial learning to recompose human-object pairs and interaction between different HOI instances as new training samples. In this way they promote knowledge generalization across HOI classes and form new triplets to mitigate the long-tail problem.

Although these methods have obtained great progress, they have not fundamentally addressed the long-tail issue caused by data imbalance. Recently, Artificial Intelligence Generated Content (AIGC) has made significant advancements in generating highly realistic images. This has opened up possibilities for using AI-generated data as a replacement for real data. Therefore, a straightforward idea is to utilize generative models to synthesize rare categories in the HOI dataset. Unfortunately, as shown in Fig. 1 (a), we find that simply merging the generated images into the original dataset and training the model does not significantly improve the performance of the model. [4, 40] have made some initial explorations about this, they alleviated the problem by designing different rules to filter out low-quality images that exhibit substantial deviation from

the original dataset. However, these approaches lack scalability across different tasks and suffer from large human subjectivity. In order to explore the critical reasons, as depicted in Fig. 1 (b), we use t-SNE[31] to analyze the feature distribution between original data and generated data. From the visualization results, it can be found that there are significant distribution differences between the generated data and the original data. In other words, there exists a considerable domain gap between the original and generated data.

To address the aforementioned issues, we propose a novel yet simple feature alignment framework called **C**ontext-**E**nhanced **F**eature **A**lignment (CEFA) module, which aims to align the generated data with the original data at the feature level and bridge the domain gap. On one hand, most existing feature alignment methods based on transformer architecture [7, 14, 36, 47] mainly align encoder and decoder features through the discriminator, without considering the instance information. While in the Human-Object Interaction tasks, the instance information of human-object pairs is critical for successful task execution. Since HOI triplets are predicted through the features of the decoder, we aim to pay more attention to instance alignment during the decoder feature alignment process. Considering that tokens in the decoder can often be repetitive or even erroneous, we propose a graph-based prototype instance alignment method. This approach views the high-scoring tokens as prototypes and constructs a graph network to align the human-object pairs by aggregating instance information. On the other hand, some researches [10, 21] have indicated that the above discriminator-style alignment method inevitably leads to losing important context clues. These context clues provide important contextual information to identify human-object interactions. For example, when the background is snow, the model should be more inclined to predict "people holding skis". Based on the above insights, we propose a context-enhanced image reconstruction module, aiming to improve the model's learning ability of contextual cues. Benefitting from the above two modules, as shown in Fig. 1 (a) and (c), our model effectively extracts more domain-independent features and bridges the domain gap.

In summary, the primary contributions of our paper are as follows:

- We propose a novel model-agnostic method CEFA which can effectively bridge the domain gap between the generated and original domains.
- We build an instance feature alignment module to align the human-object pairs by aggregating instance information. Meanwhile, we design a context-enhanced image reconstruction module to improve the model's learning ability of contextual cues.
- The proposed CEFA is a plug-and-play module that can be conveniently applied to existing HOI models and boost the performance of rare categories.

## 2 Related Work

*Long tail solution in HOI detection.* To address long-tail phenomena in HOI, researchers have proposed various methods [2, 12, 15, 34, 38, 54], which can be summarized into two categories: training strategy-based methods and combinatorial learning-based methods. Training strategy-based methods [11, 33, 44] primarily improve the

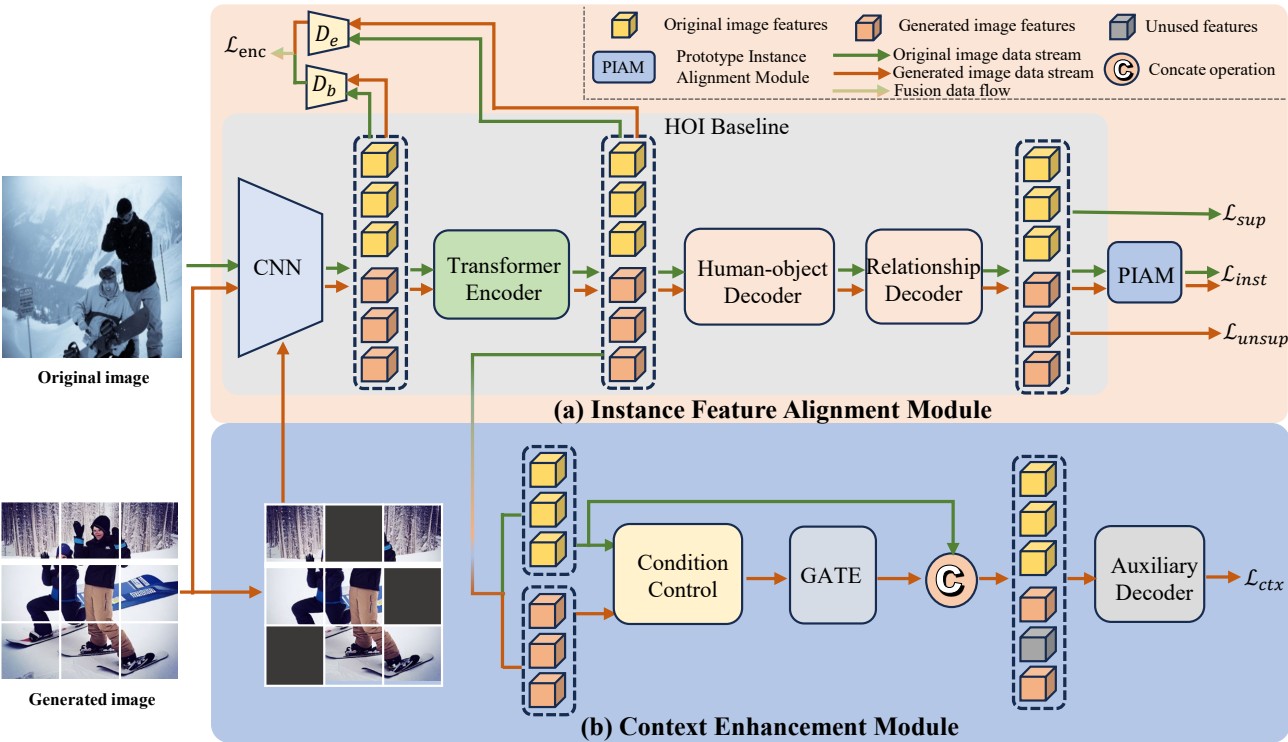

**Figure 2: The architecture for our CEFA. It consists of two parts: (I.) Instance Feature Alignment Module (orange part) : This module aligns the human-object pairs by aggregating instance information. (II.) Context Enhancement Module(blue part) : This module uses a context-enhanced image reconstruction module to improve the model's learning ability of contextual cues. The gray part in the figure represents the baseline model of HOI. Further details about PIAM can be found in Fig. 3.**

training process. They attempt to alleviate the long-tail problem by adjusting loss functions, sample sampling strategies, class balancing techniques, etc. Combinatorial learning-based methods [13, 35, 55] aim to enhance the performance of HOI models by recombining different representations of human-object pairs and interaction, forming new triplets to mitigate the long-tail problem. While existing methods have not fundamentally addressed the long-tail issue caused by data imbalance. AIGC offers a promising solution by using generative models to synthesize rare classes in the HOI dataset. Recently [4, 40] has made some initial explorations, while they suffer from the problem of large differences between the generated images and the original images. They addressed this by devising a set of rules to filter the generated images. These approaches lack scalability across different tasks and have relatively large human subjectivity. To this end, we propose CEFA to align the generated data with the original data at the feature level.

*Unsupervised domain adaptation.* To enhance the domain adaptation capabilities of the model, researchers have proposed various unsupervised domain adaptation techniques [5, 17, 42, 49]. Next, we will introduce Unsupervised domain adaptation using object detection as an example. For object detection, the goal of unsupervised domain adaptation is to narrow the domain gap between the source and target domains so that the source data can be utilized to train

a better detector for the target data. Most existing domain adaptive detectors [7, 14, 26, 36, 47] employ a teacher-student model to generate pseudo-labels and utilize adversarial learning to encourage the model to extract domain-invariant features, thereby improving domain transferability. Among them, based on the transformer architecture [7, 14, 27, 36, 47] primarily align feature extracted by CNN and token sequences from the transformer encoder-decoder. For instance, [36] aligns features at global and local levels to enhance performance in domain-adaptive object detection. While recent studies [10] have shown that the above discriminator-style alignment method will cause the model to focus on domain-invariant features while losing important context clues. We propose a context enhancement module to alleviate this problem.

## 3 Propose Method

In this section, we introduce our **C**ontext-**E**nhanced **F**eature **A**lignment (CEFA) framework, which aims to reduce the distribution difference between the generated domain and the original domain, thereby alleviating the HOI long-tail problem. As shown in Fig. 2, the CEFA comprises an "Instance Feature Alignment Module" and a "Context Enhancement Module".

Since our CEFA is a plug-and-play paradigm that can be applied to the previous one-stage based HOI model, we begin with a detailed introduction to the foundational architecture of the HOI

model in Section 3.1. Subsequently, in Section 3.2, we expound on the instance feature alignment module, which aligns the human-object pairs by aggregating instance information. In Section 3.3, we describe the context enhancement module, which employs a masked image reconstruction branch to strengthen the model's ability to capture contextual cues. Finally, in Section 3.4, we provide a comprehensive introduction to the loss function.

## 3.1 HOI Model Overview

In this section, we first introduce the formal definition of the HOI detection problem. Given a human-object image $I$, the model is required to predict a set of HOI triplets $S = \{(b_i^h, b_i^o, v_i), i \in \{1, 2, ..., N\}\}$, where $b_i^h$, $b_i^o$ and $v_i$ denotes a human bounding-box, an object bounding-box and their corresponding action category, respectively. Since our CEFA is a plug-and-play paradigm, we would first introduce the basic HOI model (e.g., [44]).

A basic HOI model can be divided into three components: visual feature extractor, human-object pair decoder, and relationship decoder. The visual feature extractor, consisting of a CNN and a Transformer encoder, is responsible for extracting image features and outputting encoder features $X_e \in R^{(H' \times W') \times D_c}$. The encoder features $X_e$ and the human-object pair query $Q_{ins} \in R^{N_q \times D_{hq}}$ are fed into the human-object pair decoder to output $X_{ho} \in R^{N_q \times D_{hd}}$, which is used to detect the position of humans and objects. Then the encoder features $X_e$, the human-object pair decoder outputs $X_{ho}$ are fed into the relationship decoder. The relationship decoder outputs $X_{ve} \in R^{N_q \times D_{ve}}$ to detect relationship categories for the corresponding pairs of humans and objects.

Although the above basic pipeline has achieved promising results in HOI detection. Unfortunately, due to the natural bias from the real world, existing methods mostly struggle with rare human-object pairs and lead to sub-optimal results. Recently, with the development of the generative model, a straightforward approach is to build a more balanced dataset with a group of supplementary samples. However, as shown in Fig. 1 (a), we find that simply merging the generated images into the original dataset cannot significantly improve the performance of the model. In order to reveal the potential reasons, as depicted in Fig. 1 (b), we use t-SNE[31] to analyze the distribution between original data and generated data. From the visualization results, it can be found that there exists a considerable domain gap between the original and generated data. To bridge the domain gap and align features, we propose the instance feature alignment module and the context enhancement module. Next, we will provide a detailed introduction to these modules.

## 3.2 Instance Feature Alignment Module

Many previous domain alignment techniques[7, 14, 36, 47] based on the transformer architecture primarily align features extracted by CNN and token sequences from the transformer encoder-decoder. Most of them feed both CNN features and all tokens into discriminators for alignment. For CNN features and transformer encoder features, similar as in SFA[36], we feed them into the discriminator for alignment, with the following loss:

$$L_{enc} = E_{X_b \in src} \log D_b(X_b) + E_{X_b \in gen} \log(1 - D_b(X_b)) \\ + E_{X_e \in src} \log D_e(X_e) + E_{X_e \in gen} \log(1 - D_e(X_e)), \quad (1)$$

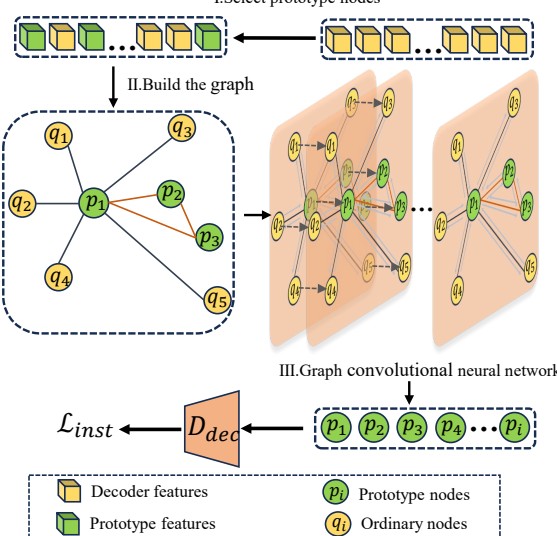

**P**rototype **I**nstance **A**lignment **M**odule(**PIAM**)

**Figure 3: The architecture of Prototype Instance Alignment Module. It consists of three steps. First, prototype selection is performed, then a graph is constructed based on prototype features and common features. Finally, the graph convolutional neural network is used to aggregate instance information.**

where $src$ represents the images from the original domain and $gen$ represents the images from the generated domain, $D_b$ and $D_e$ denotes the domain discriminator. $X_b$ and $X_e$ denotes the CNN features and transformer encoder features.

In the Human-Object Interaction tasks, the instance information of human-object pairs is critical for successful task execution. Since HOI triplets are predicted through the features of the relationship decoder, we aim to concentrate more on instance alignment during the relationship decoder feature alignment process. Considering that tokens in the relationship decoder can often be repetitive or even erroneous, feeding all relationship decoder tokens into the discriminator for alignment would lead to suboptimal results. Therefore, as shown in Fig. 3, we propose a prototype instance alignment module to aggregate instance information.

For the output of the relationship decoder, due to most of the predictions learned by the token are redundant or even incorrect, we select the top $k$ sets of relationship tokens with the highest scores as human-object instances. Following [53], each encoding can be viewed as a prototype, we consider the selected instances as prototypes.

Considering the presence of redundant tokens, in order to better aggregate instance information, we aim to enable regular tokens to transfer instance information into the prototypes. Given the convenience of graph networks in specifying information propagation rules, we construct a graph network to aggregate instance information. Specifically, we construct a graph $\mathcal{G} = (\mathcal{V}, A, X)$, where $\mathcal{V} = \{v_0, v_1, \ldots, v_{N_q}\}$ represents the set of $N$ nodes, $A \in$

$\mathcal{R}^{N \times N}$ represents the adjacency matrix of the graph, and $X \in \mathcal{R}^{N \times D_{ve}}$ represents the node feature matrix. Each graph node represents a token feature. We classify the nodes representing prototypes as prototype nodes, while the rest are referred to as regular nodes. To propagate information from regular nodes to prototype nodes for instance aggregation, there are edges connecting prototype nodes with regular nodes and other prototype nodes. There are no edges connecting regular nodes to regular nodes.

The Graph Convolutional Network (GCN) [46] can effectively capture the local and global context of nodes in the graph. By facilitating information transfer between nodes, it enables effective representation learning. So we utilize GCN for feature propagation among the nodes. Following [46], the formula for the feature propagation is as follows:

$$X = \text{GCN}\,(A, X)\,, \tag{2}$$

$$X_p = \{X_i | i \in P\}\,, \tag{3}$$

where $A$ represents the adjacency matrix, $X$ represents the node feature matrix, $P$ represents the set of prototype indices.

We extract the features of the prototype nodes as the instance features $X_p$ and input them into the discriminator for feature alignment. The instance loss is defined as follows:

$$L_{inst} = E_{X_p \in src} \log D_{dec}\,(X_p) + E_{X_p \in gen} \log\,(1 - D_{dec}\,(X_p))\,, \tag{4}$$

where $D_{dec}$ denotes the domain discriminator.

In summary, the adversarial optimization objective is formulated:

$$L_{adv} = \max_T \min_D (L_{inst} + L_{enc})\,, \tag{5}$$

where $T$, $D$ denotes the HOI detector and discriminators respectively. Gradient Reverse Layers(GRL)[5] is adopted for min-max optimization.

## 3.3 Context Enhancement Module

The aforementioned feature alignment module aligns features from three different sources: CNN features, transformer encoder features and instance features. While studies[10, 21] indicate that the above discriminator-style alignment method will cause the model to focus on domain-invariant features at the expense of losing important context clues. In the HOI task, context clues can provide rich scene information to help the model better identify human-object interactions. So we propose a context-enhanced image reconstruction module to improve the model's learning ability of contextual cues.

As shown in Fig. 2, we randomly mask the generated image $I_{gen}$ at a certain ratio $\sigma$ to obtain $I_{mask}$. The original image $I_{src}$ and the masked generated image $I_{mask}$ are separately fed into the encoder to obtain the features $X_{src}$ and $X_{mask}$. Considering that the generated images share the same semantic information as the original images with the same labels, we use the features of the original images as auxiliary information to aid the recovery process of the masked images. Firstly, we fed $X_{src}$ and $X_{mask}$ into the *conditional controller*. The role of the *conditional controller* is to learn the importance of different features from the original image for the image restoration process. It is composed of the decoder layer from the Transformer [32]. During this process, the $X_{src}$ serves as the query as well as $X_{mask}$ serves as the key and value. The output of *conditional controller* is denoted as $D_{sign}$. Secondly, Considering that not all features in the original image contribute

to the recovery of the generated image, we select the $D_{sign}$ that are greater than 0.5 using a *gate* mechanism. The final original image feature used for conditionally control $X_{src\_cond}$ is obtained by element-wise multiplication between $X_{src}$ and filtered $D_{sign}$. In summary, the *conditional controller* and the *gate* mechanism work together to filter out the feature tokens irrelevant to image restoration, extracting only the useful features that facilitate the recovery of the image. The above computations can be formulated as:

$$D_{sign} = \text{Condition}(X_{src}, X_{mask})\,, \tag{6}$$

$$\text{Sign} = \text{Gate}(D_{sign})\,, \tag{7}$$

$$X_{src\_cond} = \text{Sign} \odot X_{src}\,, \tag{8}$$

where Condition and Gate are conditional controller and gate, $\odot$ indicates the element-wise multiplication.

We concatenate $X_{src\_cond}$ with $X_{mask}$ to obtain the fused features. These fused features are then fed into an auxiliary decoder similar to MAE [9] to reconstruct the original image. It can be formulated as:

$$I_{recon} = \text{Decoder}(\text{Concate}(X_{src\_cond}, X_{mask}))\,. \tag{9}$$

Following [9], our context-enhanced image reconstruction loss $L_{ctx}$ is calculated by computing the Mean Squared Error(MSE) loss between the generated image and the reconstructed image. The calculation formula is as follows:

$$L_{ctx} = \frac{1}{n} \sum_{i=1}^{n} (I_{recon}^i - I_{gen}^i)^2\,, \tag{10}$$

where $I_{recon}^i$ represents the reconstructed image patch and $I_{gen}^i$ represents the generated image patch.

## 3.4 Loss

Our model has two types of inputs: generated images and original images. We denote $L_{sup}$ as the supervised HOI loss of original images and $L_{unsup}$ as the HOI loss of generated images. $L_{sup}$ is computed using labeled original images and $L_{unsup}$ is calculated using generated images with pseudo-labels.

The loss for the original images $L_{src}$ includes the supervised HOI loss and the discriminator loss, defined as:

$$L_{src} = L_{sup} + L_{adv}\,. \tag{11}$$

The loss for the generated images $L_{gen}$ includes the HOI loss, the discriminator loss, and the context reconstruction loss, defined as:

$$L_{gen} = L_{unsup} + L_{ctx} + L_{adv}\,. \tag{12}$$

Therefore, the total loss is given by:

$$L_{loss} = L_{src} + L_{gen}\,. \tag{13}$$

## 4 Experiments

## 4.1 Experimental Setting

*Datasets.* Following previous work, we evaluated our model on two common benchmarks, HICO-Det [3] and V-COCO [8]. HICO-Det consists of 47,776 images, with 38,118 images used for training and 9,658 images used for testing. It includes 80 object categories and 117 action categories, forming a total of 600 HOI triplets. Among these 600 HOI categories, any category with fewer than 10 training instances is defined as Rare, while the remaining categories are

**Table 1: Comparisons with the state-of-the-art methods on the HICO-Det datasets. We report Full, Rare, and Non-rare accuracies. †represents a model trained directly on generated data, without using our CEFA method.**

| Method | Detector | Backbone | Default | | | Known Object | | |
|---|---|---|---|---|---|---|---|---|
| | | | Full | Rare | Non-rare | Full | Rare | Non-rare |
| DRG[6] | HICO-DET | ResNet50-FPN | 24.53 | 19.47 | 26.04 | 27.98 | 23.11 | 29.43 |
| GG-Net[51] | HICO-DET | Hourglass104 | 23.47 | 16.48 | 25.60 | 27.36 | 20.23 | 29.48 |
| IDN [22] | HICO-DET | ResNet50 | 26.29 | 22.61 | 27.39 | 28.24 | 24.47 | 29.37 |
| QPIC[30] | HICO-DET | ResNet50 | 29.07 | 21.85 | 31.23 | 31.68 | 24.14 | 33.93 |
| SCG [45] | HICO-DET | ResNet50-FPN | 31.33 | 24.72 | 33.31 | 34.37 | 27.18 | 36.52 |
| DT [52] | HICO-DET | ResNet50 | 31.75 | 27.45 | 33.03 | 34.50 | 30.13 | 35.81 |
| STIP [48] | HICO-DET | ResNet50 | 31.60 | 27.75 | 32.75 | 34.41 | 30.12 | 35.69 |
| HQM [50] | HICO-DET | ResNet50 | 32.47 | 28.15 | 33.76 | - | - | - |
| MSTR [19] | HICO-DET | ResNet50 | 31.17 | 25.31 | 32.92 | 34.02 | 28.83 | 35.57 |
| RLIP [43] | COCO+VG | ResNet50 | 32.84 | 26.85 | 34.63 | - | - | - |
| IF [24] | HICO-DET | ResNet50 | 33.51 | 30.30 | 34.46 | 36.28 | 33.16 | 37.21 |
| CDN[44] | HICO-DET | ResNet50 | 31.44 | 27.39 | 32.64 | 34.09 | 29.63 | 35.42 |
| CDN† | HICO-DET | ResNet50 | 31.41 | 27.27 | 31.42 | 34.02 | 27.27 | 34.05 |
| CDN+*Ours* | HICO-DET | ResNet50 | 31.86 | 28.50(+1.11) | 32.87 | 34.51 | 31.00(+1.37) | 35.57 |
| GEN-VLKT [23] | HICO-DET | ResNet50 | 33.75 | 29.25 | 35.10 | 36.78 | 32.75 | 37.99 |
| GEN-VLKT† | HICO-DET | ResNet50 | 33.74 | 29.30 | 35.04 | 36.67 | 32.71 | 37.86 |
| GEN-VLKT+*Ours* | HICO-DET | ResNet50 | 34.19 | 30.74(+1.49) | 35.23 | 37.04 | 33.65(+0.9) | 38.05 |
| HOICLIP [25] | HICO-DET | ResNet50 | 34.57 | 30.71 | 35.70 | 37.96 | 34.49 | 39.02 |
| HOICLIP† | HICO-DET | ResNet50 | 34.46 | 30.70 | 35.59 | 37.58 | 34.02 | 38.64 |
| HOICLIP+*Ours* | HICO-DET | ResNet50 | 35.00 | 32.30(+1.59) | 35.81 | 38.23 | 35.62(+1.13) | 39.02 |

**Table 2: Comparisons with the state-of-the-art methods on the V-COCO datasets. We report Full(S1), Rare, and Non-rare accuracies. * signifies results reproduced with the official implementation codes.**

| Method | Backbone | Full(S1) | Rare | Non-rare |
|---|---|---|---|---|
| HOTR[18] | R50 | 55.20 | 39.71 | 57.67 |
| QPIC[30] | R50 | 58.80 | 42.12 | 60.87 |
| CDN[44] | R50 | 61.68 | 45.98 | 65.01 |
| CDN+*Ours* | R50 | 62.70 | 48.21(+2.23) | 65.02 |
| GEN-VLKT [23] | R50 | 62.41 | 48.16 | 65.23 |
| GEN-VLKT+*Ours* | R50 | 63.15 | 49.80(+1.64) | 65.29 |
| HOICLIP* [25] | R50 | 63.23 | 50.38 | 65.28 |
| HOICLIP+*Ours* | R50 | 63.53 | 51.76(+1.38) | 65.41 |

defined as Non-Rare. V-COCO is a subset of the COCO dataset and comprises 10,396 images, with 5,400 images used for training and 4,964 images used for testing. It contains 29 action categories, including 4 body actions that do not involve interactions with any objects. The dataset shares the same 80 object categories as HICO-Det. The actions and objects together form 263 HOI triplets. Among the 29 action categories, we define action categories with less than 300 training instances as Rare, and the remaining action categories as Non-Rare.

*Implementation details.* We use Blip-diffusion [20] to generate images with the template "a photo of human verbing object", where verbing represents the present progressive form of the action in the HOI triplet. The threshold for pseudo-label with the model is set to 1.4 for GEN-VLKT and HOICLIP benchmark models, and 0.4 for the CDN benchmark model. During the training phase, we fine-tune the pre-trained weights of the existing HOI model and keep the human-object pair decoder fixed. The learning rate for the backbone in the CDN benchmark is set to 1e-5, while the rest of the learning rates are set to 1e-4. For the GEN-VLKT and HOICLIP benchmark, the backbone learning rate is set to 1e-6, and the rest is set to 1e-4. The mask ratio $\sigma$ is 0.8, and the number of prototype nodes $k$ is set to 6. We conduct all the experiments on 8 Tesla V100 GPUs, using a batch size of 12 per GPU.

*Evaluation.* We evaluated our model using mean Average Precision (mAP). The HOI triplet predictions are considered true positives when they meet the following criteria: 1) the predicted human and object bounding boxes have an IoU greater than 0.5; and 2) both predicted HOI categories are accurate. For HICO-Det, we evaluated three different category sets: the complete set of 600 HOI categories (Full), the 138 HOI categories with fewer than 10 training instances (Rare), and the remaining 462 HOI categories (Non-Rare). For V-COCO, we also evaluated three different category sets: the complete set of 29 action categories (Full), the 4 action categories with fewer than 300 training instances (Rare), and the remaining 25 action categories (Non-Rare).

## 4.2 Quantitative Evaluation

To evaluate the effectiveness of our method, we conducted combined experiments with three representative methods. Firstly, we

**Table 3: Ablation studies on the HICO-Det dataset. "Instance" represents the instance features alignment module. "Context" represents the context enhancement module.**

|      | Instance | Context | Full | Rare | Non-Rare |
|------|----------|---------|------|------|----------|
| 1    |          |         | 34.57 | 30.71 | 35.70 |
| 2    | ✓        |         | 34.89 | 31.77 | 35.79 |
| 3    |          | ✓       | 34.92 | 31.91 | 35.80 |
| Ours | ✓        | ✓       | **35.00** | **32.30** | **35.81** |

**Table 4: The effects of different model settings. More detailed discussion can be found in Sec. 4.4.**

|   |                                              | Full  | Rare  | Non-Rare |
|---|----------------------------------------------|-------|-------|----------|
| 1 | Context:Non-Condition                        | 34.86 | 31.83 | 35.75    |
| 2 | Instance:Graph → Transformer                 | 34.75 | 31.29 | 35.79    |
| 3 | Control:Cross-attention → MLP                | 34.76 | 31.54 | 35.75    |
| 4 | Control:Cross-attention → Self-attention     | 34.93 | 32.11 | 35.78    |
| 5 | Graph:bidirectional→ fully connected         | 34.59 | 31.38 | 35.71    |
| 6 | Graph:bidirectional→ directed                | 34.92 | 32.00 | 35.79    |
| 7 | Ours                                         | **35.00** | **32.30** | **35.81** |

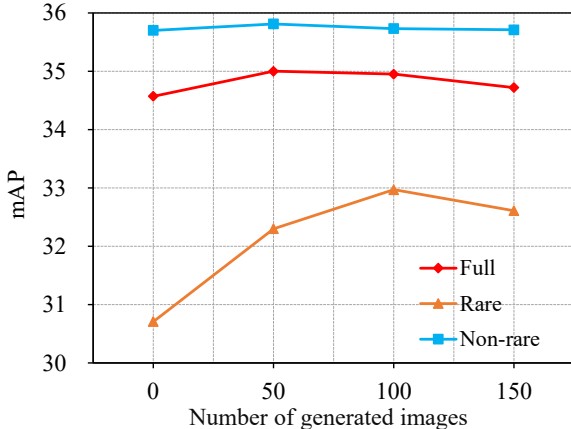

**Figure 4: The effects of different numbers of generated images for per rare class on the HICO-Det dataset.**

performed experiments on the HICO-Det dataset, the results are shown in Table 1. Since the proposed method is a plug-and-play module, our method is combined with CDN, GEN-VLKT, and HOICLIP to achieve performance improvements. Specifically, when combined with CDN, our method achieved a relative improvement of 4.05% on rare categories. When combined with GEN-VLKT and HOICLIP, our method achieved a +1.49 mAP and +1.59 mAP improvement on rare categories. These results demonstrate the effectiveness of our method in handling rare class HOI tasks. We have also noted that merely merging the generated data with the original dataset for training does not lead to a direct and intuitive improvement in performance. This confirms that the domain gap between the original and generated data causes the model to deviate from the optimal solution during the fine-tuning process, thereby impeding the enhancement of performance. After applying our method, there is a significant increase in performance, which further attests to the effectiveness of our approach. Additionally, the compatibility of our method with various other methods also proves its excellent extensibility. We also compared our method to previous state-of-the-art (SOTA) approaches, it can be found that our method outperforms these works by a big margin.

Next, Table 2 presents the performance of our method on the V-COCO dataset. The results showed significant improvements over the original models. Specifically, when CDN is combined with our method, we achieved a +1.02 mAP improvement for all categories and an impressive +2.23 mAP improvement on rare categories. When combined with GEN-VLKT and HOICLIP, we achieved a +1.64 mAP and +1.38 mAP improvement on rare categories.

## 4.3 Ablation Studies

*Model components ablation.* In this section, we conduct several ablation studies on the HICO-Det to systematically evaluate the

contributions of different model components. As shown in Table 3, the "Instance" and "Context" represent instance feature alignment module and context enhancement module, respectively. Specifically, "Instance" indicates the process of aligning human-object pairs by aggregating instance information. "Context" represents using context-enhanced image reconstruction module to improve the model's learning ability of contextual cues. Due to HOICLIP [25] achieving state-of-the-art results in previous methods, we selected it as the baseline for comparison. As shown in rows 1-2 of Tabel 3, we follow the baseline and add an instance feature alignment module to align features. We can observe that the rare class mAP achieves an improvement of 1.06, which demonstrates that our instance feature alignment module is capable of effectively aligning human-object pairs, thereby promoting domain alignment. In addition, in row 3 of Table 3, we follow the baseline and add the context enhancement module separately to extract important contextual information. The context enhancement module resulted in a relative improvement of 3.91% on rare categories, which shows that the proposed context enhancement module can effectively learn important contextual cues and alleviate the loss of contextual information.

## 4.4 Alternative Experiments

*Alternative model settings.* In Table 4, we exploit some alternative experiments about the different model settings on the HICO-Det dataset. To validate the effectiveness of our proposed conditional controller in the context enhancement module, we removed it and used the original mask autoencoder to reconstruct the images. The results in the first row indicate that adding the original image as a conditional control can further guide the model to better reconstruct images and promote the learning of contextual clues. In the second row of the table, we replaced the graph convolutional network with a transformer model in the instance feature alignment module. The results show that the transformer model is not well-suited for aggregating instance information. The main reason is that the transformer model does not consider the special relationship between prototype nodes and regular nodes.

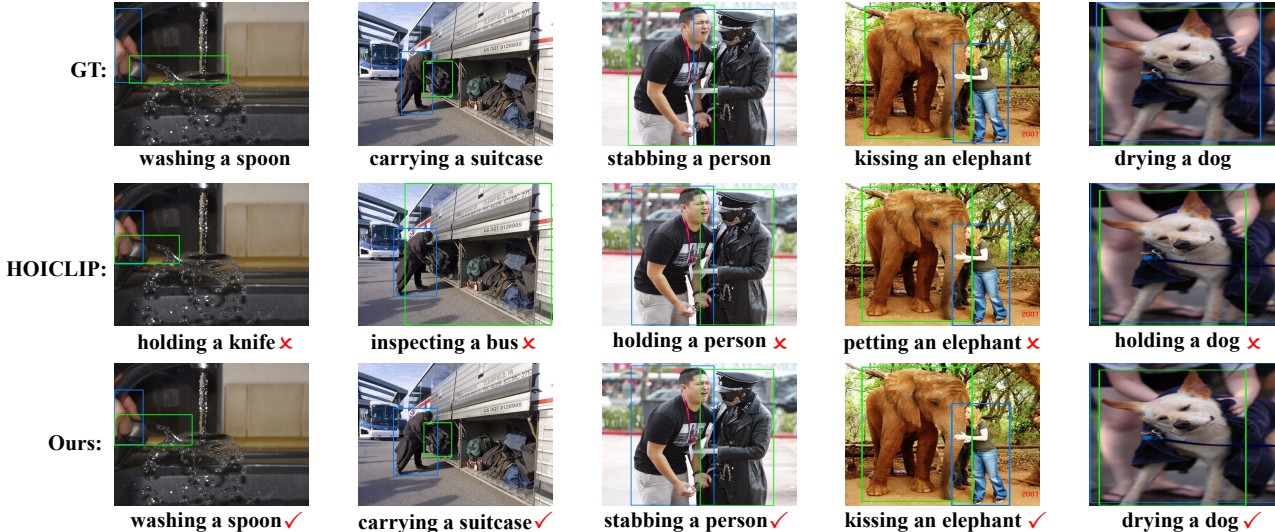

**Figure 5: Qualitative evaluation on the HICO-Det dataset. The first row represents the ground truth, the second row shows the predictions from the previous benchmark model HOICLIP, and the third row presents the results from our proposed method. In the figure, blue bounding boxes represent person, and green bounding boxes represent objects.**

In the 3-4 rows of Table 4, we replaced the cross-attention with MLP and self-attention in the conditional controller, respectively. The results indicate that MLP cannot effectively serve as the conditional controller because it lacks the ability to fuse and control information compared to cross-attention. We observed that using both cross-attention mechanisms and self-attention mechanisms as conditional controllers yielded excellent performance, which proves that these two mechanisms can extract meaningful conditional control signals.

As shown in rows 5-6 of Table 4, we replaced the bidirectional graph with fully connected and directed graphs. Directed graphs refer to there are edges connecting from ordinary nodes to prototype nodes. In row 5 of Table 4, our bidirectional graph outperformed the fully connected graph by 0.92 mAP, indicating the importance of considering the special relationships between prototype and ordinary nodes. Row 6 indicates a slight impact of relationship propagation from prototype nodes to ordinary nodes.

*Generate quantity settings.* As shown in Fig. 4, we studied the impact of the number of generated images on the HICO-Det. We find that the performance of rare classes improves with an increasing number of generated images, but does not increase after the number of generated images exceeds 100. In order to maintain a balance between performance and resource consumption, we decided to generate 50 images per rare class.

### 4.5 Qualitative Evaluation

To more intuitively demonstrate the effectiveness of our approach, we visualize some examples of experimental results, as shown in Fig. 5. In the first row, we show the ground truth labels. The second row displays the predictions of the previous state-of-the-art method HOICLIP[25]. The third row represents the predictions of our CEFA

model. In the figure, blue bounding boxes represent person, and green bounding boxes represent objects.

By observing the 1-2 columns, we find that the HOICLIP exhibits errors in the localization and classification of humans and objects, which subsequently affects the prediction of relationship interactions. In contrast, our model achieves more accurate localization and classification, enabling the correct prediction of relationship interactions. Furthermore, by observing the the last three columns, we can observe that even the HOICLIP correctly detects humans and objects, errors can still occur due to failed relationship predictions. While our model is capable of improving this situation.

### 4.6 Visualization of Feature Distribution

We present the distribution of features extracted by HOICLIP. As shown in Fig. 1 (b), the generated image and original image features can be easily separated by domain. In contrast, our proposed CEFA model is capable of learning domain-invariant features during the feature extraction phase. As illustrated in Fig. 1 (c), after processing with our approach, the image features from different domains become significantly more aligned and are not easily distinguishable.

### 5 Conclusion

In this paper, we introduce a model-agnostic framework that can be conveniently applied to existing HOI models and boost the performance of rare categories. Specifically, we propose **C**ontext-**E**nhanced **F**eature **A**lignment (CEFA) module to align the generated data with the original data at the feature level. The CEFA helps the model overcome the domain gap, allowing the use of generated data for practical training. Experimental results demonstrate that our CEFA achieves promising performance in terms of accuracy and scalability. We expect that this approach will provide a new paradigm for our community.

## Acknowledgments

This work was supported by the National Natural Science Foundation of China (No.62102323), the National Natural Science Foundation of China (U23B2013) and the Innovation Capability Support Program of Shaanxi(Program No. 2023KJXX-142).

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
