# OpenReview forum: "A Plug-and-Play Method for Rare Human-Object Interactions Detection by Bridging Domain Gap"
_acmmm.org/ACMMM/2024/Conference — MM2024 Poster_

### Official Review · Reviewer_DCSw · 2024-05-23

**Rating:** 4
**Confidence:** 3

**Summary:**

This work addressed the domain gap (between generated images and training set) of human-object interaction detection, especially for the rare classes. A model-agnostic framework to align the generated data with the original training set a the feature level. The model is plug-and-play and the improvement compared to naive data augmentation is of good margin.

**Strengths:**

+ This work leverages the generative AI model to augment training data for HOI detection tasks. The authors observed that naive augmentation does not work., due to image quality and domain gap.
+ A novel framework (i.e., CEFA) was proposed to perform feature-level alignment to bridge the domain gap, which considers the instance alignment through a graph-based prototype instance alignment method. In addition, a context-enhanced image reconstruction module is proposed for learning the contextual cues.
+ The proposed CEFA is model agnostic and can benefit many HOI detection approaches.
+ The proposed CEFA shows good improvement on rare classes on three HOI baselines. The authors also provide a number of ablation studies to support the claims.

**Limitations:**

- Can the author provide an intuition of the context enhancement module? Specifically, how can the proposed method leverage more context clues using the proposed mechanism?
- For the ablation studies, generate quantity settings, is the number of generated images constant for all classes? Has the author considered a more class-dependent strategy where more images are generated for rare classes and less for the common classes?
- In supplementary material, Section 4, it indicates that the negative predictions are mainly attributed to the incorrect localisation and classification of human-object pairs. This is a STRONG statement, can you author provide evidence that if ground truth object localisation and class label are given, the model can achieve highly accurate results?
- Tab 3 shows that both the instance feature alignment module and context enhancement module improve the overall HOI detection performance. In this work, the generated samples are fixed and no steps are performed to improve the generated image quality (or diversity). I am curious about which of the modules can better reduce the bias of the learned features that are shown in Fig 1(c).

Others
- In Fig 1's caption, "baseline+Date" --> "Baseline+Data"

**Suitability:**

2

---

### Official Review · Reviewer_EyFU · 2024-05-24

**Rating:** 4
**Confidence:** 3

**Summary:**

This work tackles the human-object interaction (HOI) detection task on 2d images, which is essentially an image-based unimodal detection task similar to object detection. As multi-modal diffusion models are adopted in the proposed method to enhance the performance of the HOI detection model, I think the topic of this work is moderately (but not strongly) related to the MM society.

This work use the generated images from diffusion models to augment the HOI detector, especially on the tail classes with fewer natural samples. To tackle the significant domain gap between the generated images and the original samples, this work proposed a method with two components. The first is to align the instance features from the original samples and the generated samples. The second is to reconstruct the generated samples from image patches, to strengthen the learning on contextual cues.

**Strengths:**

- The major problem to tackle in this work is valid and somewhat interesting. Two previous works try to benefit HOI detection models with generated samples, but they do not notice the domain gap between the generated and the original ones, which is empirically a trouble for this approach.
- Moderate experimental improvement on rare (tail) classes. This shows the method is effective on the tail classes, as they claim to solve.

**Limitations:**

- The relationship between the motivation and the proposed modules needs more clarification.
    - The second component, the context enhancement module, seems not strongly related to the overall motivation. This work aims to tackle the problem of domain gaps between the generated and the original samples. Why does the model need to strengthen the learning on the contextual cues? This might need more clarification. If the method is to solve the problem of losing context due to the first component, then the experimental results in Table. 3 seems not supporting this, as adding the second to the first brings limited improvement over using only one component.
    - Similarly, the relationship between the first component and the motivation, though can be understood, is not very clearly stated in the paper.
- Limited novelty and importance of the technical design for the first component. The GNN-based feature fusion in PIAM has been adopted by many HOI detectors, and the difference and necessity for such design should be clarified. Besides, Table 4 shows the specific choice of the design has little effect on the overall performance.
- The proposed method is not so "plug-and-play" and claimed. Firstly, as shown by Fig.2, it is only applicable to HOI detection methods with CNN+transformer+HOI detector+relationship decoder, which constitutes only one part of the different frameworks of HOI detectors. Secondly, the method introduces multiple auxiliary losses and additional networks. This limits the ability to transplant it onto different baselines.
- Mistakes in writing. I understand that some typos or grammatical errors in the main sections are inevitable in the paper and there are no need to mention them, but this paper contains several mistakes in the *abstract*, which should be checked multiple times before submitted. For example, "Unfortunately, there is a significant domain gap between the generated data and the original data, simply merging the generated images into the original dataset cannot significantly boost the performance" is grammatically wrong. "On one hand, Considering the crucial role ..." contains obvious typos.

**Suitability:**

2

---

### Official Review · Reviewer_721n · 2024-05-25

**Rating:** 4
**Confidence:** 3

**Summary:**

The authors propose to alleviate the long-tail problem in HOI detection. Firstly, the authors found that the generated images have a distributional shift with the real world data. Then, the authors propose two modules to align their features at the instance level. The authors conduct extensive experiments on benchmark dataset. The results showing that the proposed method can indeed improve the performance on various base models, especially on tail classes.

**Strengths:**

+ The problem studied in this paper is important.

+ The observation that generated data has a different distribution with real-world data is interesting and new to this area.

+ The proposed method is well-motivated and effective as demonstrated by experiments.

+ The ablation studies provide in-depth analysis of the proposed method.

**Limitations:**

Any idea on the domain gap between real world data and generated data? Is it related to the distribution shift between HICO-DET and LAION? On the other hand, does the generated image contain only one person? Does this also contribute to the distribution shift? It would be more insightful if the authors could further discuss this.

Following the above question. I wonder is there any difference when using different generative models? Will the domain gap be different?


Some related references are missing:

[1] Lin et al. Learning from Easy to Hard Pair: Multi-step Reasoning Network for Human-Object Interaction Detection. ACM MM 23

[2] Wang et al. Distance Matters in Human-Object Interaction Detection. ACM MM 22

**Suitability:**

3

---

### Official Review · Reviewer_CqpX · 2024-05-28

**Rating:** 5
**Confidence:** 4

**Summary:**

This paper introduces a model-agonistic framework to align the domain gap between the generated data and the original data for rare HOI detection. Empirical results indicate the effectiveness of the proposed method for HOI detection.

**Strengths:**

(1) The idea of prototype instance alignment is interesting and reasonable.
(2) Experiments are extensive and demonstrate the effectiveness of the method.
(3) The paper is generally well-written and in good quality.

**Limitations:**

(1) It is not quite clear how does the generated images be produced based on the original image. How to guarantee the generated image is in the same domain of the original one? How about the quality of the generated image?

(2) In the Instance Feature Alignment Module, how does the relationship decoder model the relationships of human and object? It seems the relationship decoder only takes the human feature and object feature as inputs, without the information of interactions, which would make it difficult to model relationships between human and object.

(3) The paper focus on modeling rare HOIs, how does the proposed method deal with the challenge of rare events?

**Suitability:**

3

---

### Meta-Review · Area_Chair_Ukuu · 2024-06-28

**Recommendation:** Accept (Poster)
**Confidence:** 4

**Metareview:**

The initial reviews lean slightly and consistently towards accepting this paper. After the efficient rebuttal, two reviewers upgraded their rates from borderline accept to weak accept. The paper proposes a model-agonistic framework to align the domain gap between the generated data and the original data for rare HOI detection. It shows strong empirical results indicating the effectiveness of the proposed method. The AC thus would like to recommend it for publication.